# Mutation analysis of multiple pilomatricomas in a patient with myotonic dystrophy type 1 suggests a DM1-associated hypermutation phenotype

**Albert Rübben** [ORCID][1]*, **Renate Ursula Wahl**[1], **Thomas Eggermann**[2], **Edgar Dahl**[3,4], **Nadina Ortiz-Brüchle**[3,4‡], **Claudio Cacchi**[3‡]

**1** Department of Dermatology, RWTH Aachen University, Aachen, Germany, **2** Institute of Human Genetics, RWTH Aachen University, Aachen, Germany, **3** Institute of Pathology, RWTH Aachen University, Aachen, Germany, **4** Medical Faculty, RWTH centralized Biomaterial Bank (RWTH cBMB), Aachen, Germany

‡ These authors are joint senior authors on this work.
* albert_ruebben@post.rwth-aachen.de

**Data Availability Statement:** All relevant data are within the manuscript. Raw NGS data are available at the National Center for Biotechnology Information Sequence Read Archive (https://www.

## Abstract

Myotonic dystrophy type 1 (DM1) is an inherited neuromuscular disease which results from an expansion of repetitive DNA elements within the 3' untranslated region of the *DMPK* gene. Some patients develop multiple pilomatricomas as well as malignant tumors in other tissues. Mutations of the catenin-β gene (*CTNNB1*) could be demonstrated in most non-syndromic pilomatricomas. In order to gain insight into the molecular mechanisms which might be responsible for the occurrence of multiple pilomatricomas and cancers in patients with DM1, we have sequenced the *CTNNB1* gene of four pilomatricomas and of one pilomatrical carcinoma which developed in one patient with molecularly proven DM1 within 4 years. We further analyzed the pilomatrical tumors for microsatellite instability as well as by NGS for mutations in 161 cancer-associated genes. Somatic and independent point-mutations were detected at typical hotspot regions of *CTNNB1* (S33C, S33F, G34V, T41I) while one mutation within *CTNNB1* represented a duplication mutation (G34dup.). Pilomatricoma samples were analyzed for microsatellite instability and expression of mismatch repair proteins but no mutated microsatellites could be detected and expression of mismatch repair proteins MLH1, MSH2, MSH6, PMS2 was not perturbed. NGS analysis only revealed one heterozygous germline mutation c.8494C>T; p.(Arg2832Cys) within the ataxia telangiectasia mutated gene (*ATM*) which remained heterozygous in the pilomatrical tumors. The detection of different somatic mutations in different pilomatricomas and in the pilomatrical carcinoma as well as the observation that the patient developed multiple pilomatricomas and one pilomatrical carcinoma over a short time period strongly suggest that the patient displays a hypermutation phenotype. This hypermutability seems to be tissue and gene restricted. Simultaneous transcription of the mutated *DMPK* gene and the *CTNNB1* gene in cycling hair follicles might constitute an explanation for the observed tissue and gene specificity of hypermutability observed in DM1 patients. Elucidation of putative mechanisms responsible for hypermutability in DM1 patients requires further research.

ncbi.nlm.nih.gov/sra); accession number
PRJNA603431, SubmissionID SUB6884814.

**Funding:** The authors received no specific funding
for this work.

**Competing interests:** The authors have declared
that no competing interests exist.

## Introduction

Myotonic dystrophy type 1 (DM1, OMIM 160900) is an inherited and the most common neuromuscular disorder characterized genetically by an expansion of trinucleotide repeats within the 3' untranslated region of the *DMPK* (DM1 protein kinase) gene [1–5]. The DNA-expansion within the *DMPK* gene is considered causative for the observed muscle weakness and cardiac disease [5,6]. Besides, affected patients develop early cataract as well as insulin resistance and cognitive impairment.

Although initially hypothesized that DM1 is primarily caused by mutations that generate an amplification of CTG repeats [1], the underlying mutation driving this amplification has not been identified yet. It has, nevertheless, been speculated that the DNA mismatch-repair mechanism as well as mechanisms involved in the resolution of secondary DNA structures such as hairpins or R-loops might be implicated in some form [7–9]. In order to explain the multiple non-muscular clinical symptoms which are associated which DM1, an alternative gain-of-function-RNA hypothesis was formulated and subsequently proven. It could be demonstrated that expanded CTG-repeats within the *DMPK* gene are transcribed into RNA and that this non-translated repetitive RNA then forms aggregates with various splicing regulators, which in turn impair transcription of multiple genes in various tissues and which might also be responsible for further expansion of CTG-repeats [5,10–12].

For more than 50 years it has been known that patients with myotonic dystrophy may develop multiple pilomatricomas (synonyms: pilomatrixoma, calcifying epithelioma of Malherbe) which are benign calcifying skin tumors deriving from hair matrix cells [13–15]. Pilomatricoma is a relatively rare tumor but it represents the second most frequent skin tumor in childhood. Age distribution seems to follow a bimodal distribution with a first marked peak in the first decennium and a second small and broad increase in prevalence between 41- and 71-years of age. Non-syndromic pilomatricoma occurs mostly as a solitary lesion in the head and neck region. While the scalp is affected in childhood in only approx. 4%, pilomatricomas of the scalp are more frequent in adulthood (approx. 27%) [16–18], whereas in myotonic dystrophy, most pilomatricomas are located on the scalp [15]. In contrast to non-syndromic pilomatricoma, myotonic dystrophy-associated pilomatricoma is a disease of adulthood. The overall frequency of pilomatricoma in DM1 seems to be lower than 10% with a male predominance [19–22].

Multiple pilomatricomas have also been encountered in patients with Turner syndrome, Rubinstein-Taybi syndrome, trisomy 9, Gardner syndrome and in patients with constitutive mismatch repair deficiency (CMMR-D) [23]. A family with non-syndromic multiple pilomatricomas has been described as well [24]. Mutations of the catenin-β gene (*CTNNB1*) have been found in many analyzed non-syndromic pilomatricomas as well as in pilomatricomas associated with constitutive mismatch repair deficiency and in pilomatrical carcinomas [23,25–29]. Pilomatricomas have been described in a few patients with *APC*-mutated familial adenomatous polyposis (Gardener syndrome) but *APC*-mutations have not been reported in non-syndromic pilomatricoma [25,30]. The WNT/APC/β-catenin pathway regulates hair follicle development, hair follicle cycling, and hair growth and β-catenin is strongly expressed in the proliferating matrix cells of pilomatricoma, both in catenin-β mutated tumors and in pilomatricomas without a *CTNNB1* mutation [28]. Therefore, mutation of the *CTNNB1* most likely represents the tumor driving oncogenic event in pilomatricoma.

Besides benign pilomatricomas, patients with DM1 also have an enhanced risk of malignant tumors. The cancer risk is elevated by a factor of approx. 1.8 [31]; the most prevalent cancers affect skin, thyroid, ovary, and breast [15]. The relative cancer risk is elevated especially for testicular cancer in men, endometrial cancer and ovary cancer in women as well as brain cancer, thyroid cancer and Non-Hodgkin's lymphoma in both sexes [15,20,31,32].

Several hypotheses have been forwarded or can be formulated in order to explain the susceptibility to pilomatricoma and cancer in myotonic dystrophy patients:

1. *Direct effect of untranslated repetitive RNA on oncogene expression*–Following the gain-of-function-RNA hypothesis, Mueller and colleagues suggested in 2009 that the untranslated repetitive RNA directly enhances expression of β-catenin resulting in pilomatricomas as well as in various cancers which rely on activation of the WNT/APC/β-catenin pathway [15].

2. *Second mutation inducing genetic instability*–It has been suggested that the same cellular mechanism that allows for germline and somatic expansion of CTGn or CCTGn repeats in myotonic dystrophy patients type 1 and 2 could also lead to unchecked DNA repair errors [31]. This mechanism could be a hitherto not identified second mutation present in a subset of DM1 patients which would also explain that not all DM1 patients develop pilomatricomas or cancers.

3. *Effect of untranslated repetitive RNA on expression of genes involved in DNA proofreading or replication*–As an additional alternative, one may suggest that the untranslated repetitive RNA from the mutated *DMPK* gene interferes with the expression of genes involved in DNA proofreading and replication, thereby inducing both expansion of DNA repeats in the *DMPK* gene as well as mutations in cancer-driving genes.

4. *Direct interfering effect of untranslated repetitive RNA on molecular mechanisms involved in DNA proofreading or replication*–Alternatively to a second mutation, one may hypothesize that the untranslated repetitive RNA from the mutated *DMPK* gene directly interferes with molecular mechanisms involved in DNA proofreading or replication, for example by forming R-loops [8].

In order to gain insight into the molecular mechanisms which might be responsible for the occurrence of multiple pilomatricomas and cancers in patients with DM1, we sequenced the *CTNNB1* gene in five pilomatricomas and in one pilomatrical carcinoma from one patient with molecularly proven DM1 and further analyzed the tumors for microsatellite instability and for mutations in 161 cancer-associated genes.

## Results

### Analysis of CTG repeat expansions

The patient demonstrated one *DMPK* allele in the normal range with 5 +/- 2 repeats as well as an expanded *DMPK* allele with more than 400 CTG-repeats.

### Catenin-beta 1 gene sequencing

We analyzed the Catenin beta 1 gene (*CTNNB1*) for mutations in six samples from four benign pilomatricomas, and in one sample of a pilomatrical carcinoma obtained from the patient (Fig 1A–1E). In all samples we could detect mutations at typical hotspot regions of exon 3 of *CTNNB1* (Table 1).

Two pilomatricomas and one pilomatrical carcinoma demonstrated mutations which targeted codon 33 (S33C, S33F, S33C). One pilomatricoma demonstrating a hitherto non described small duplication within codons 33 and 34 (c.99_101dup, p.G34dup.). Another larger pilomatricoma which was microdissected at two areas demonstrated clonal heterogeneity as one area only demonstrated a mutation at codon 41 (T41I) while the other tissue area only displayed a mutation at codon 34 (G34V).

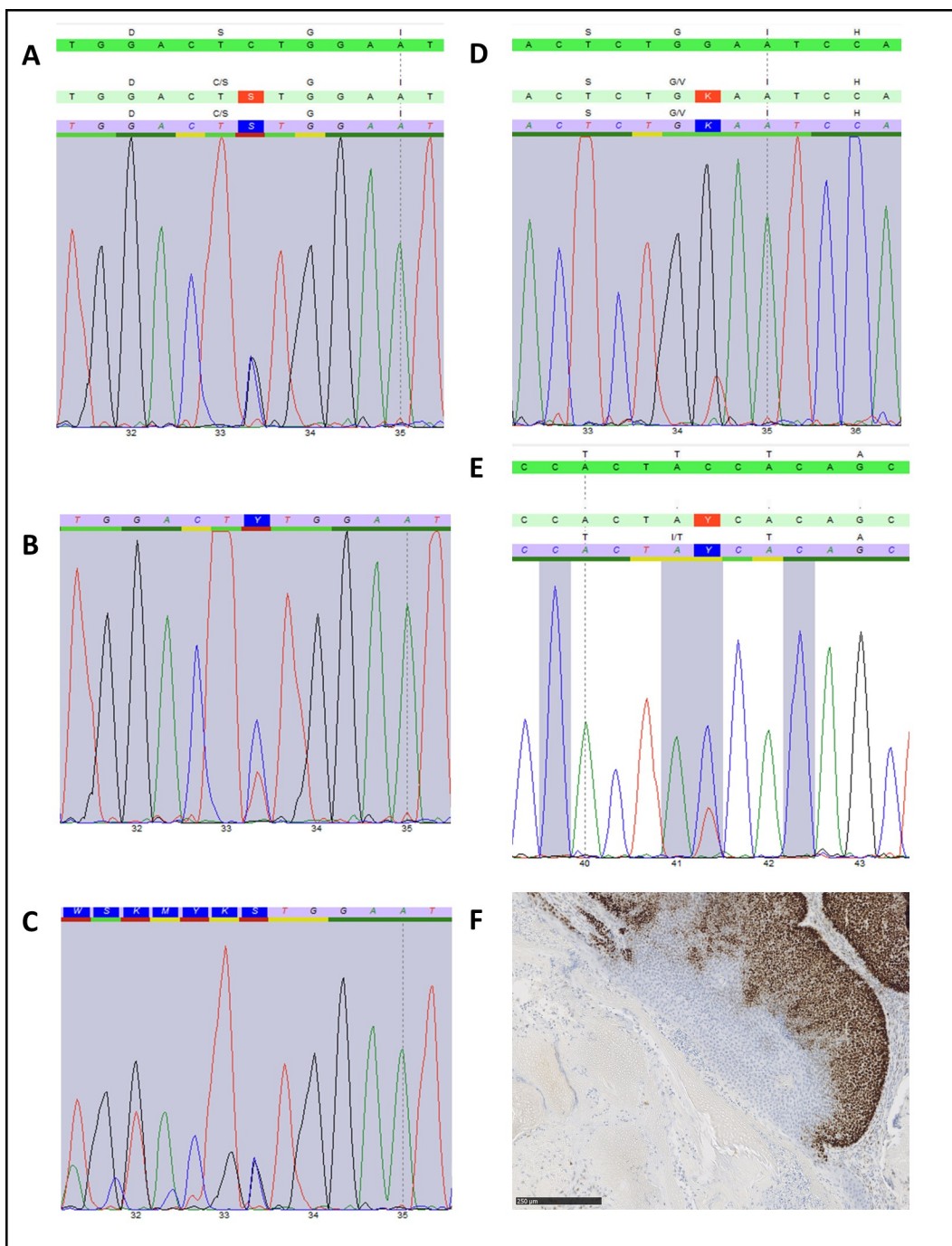

**Fig 1. *CTNNB1* sequencing.** Mutations: A > S33C, B > S33F, C > G34dup., D > G34V, E > T41I. F: Immunohistochemistry of MSH6 expression restricted to matrix cells of the pilomatricoma p.(G34dup).

Table 1 summarizes the detected mutations and compares them to published mutations in the *CTNNB1* gene found in non-syndromic and in *PMS2*-mutation associated syndromic pilomatricomas as well as in sequenced pilomatrical carcinomas [22,25–29]. This overview shows that mutations of codon 33 and 37 of *CTNNB1* are represented at about the same frequency in

**Table 1.** *CTNNB1*-mutations detected in four pilomatricomas (PM) and one pilomatrical carcinoma (PMC) of the described DM1-patient and overview of published mutations in non-syndromic and syndromic PM as well as in PMC.

| Detected mutation | | Non-syndromic PM [25–29] | CMMR-D-syndromePM [22] | PMC [29] | DM1-case (PM and PMC) |
|---|---|---|---|---|---|
| protein | Nucleotide | | | | |
| D32Y | GAC>TAC | 7 | 1 | 1 | |
| D32G | GAC>GGC | 1 | | | |
| D32V | GAC>GTC | 1 | | | |
| D32Q | GAC>CAG | 1 | | | |
| S33C | TCT>TGT | 2 | | | 1 PM, 1 PMC |
| S33F | TCT>TTT | 10 | | 3 | 1 PM |
| S33Y | TCT>TAT | 2 | | | |
| S33P | TCT>CCT | | 2 | | |
| G34R | GGA>AGA | 1 | | | |
| G34E | GGA>GAA | 3 | | 1 | |
| G34V | GGA>GTA | | | 1 | 1 PM* |
| G34dup | c99._101dup (TGGdup) ACC>ATC | | | | 1 PM |
| S37C | TCT>TGT | 3 | | 1 | |
| S37F | TCT>TTT | 3 | | 2 | |
| S37Y | TCT>TAT | 3 | | | |
| T41I | ACC>ATC | 2 | 3 | | 1 PM* |
| T41A | ACC>GCC | 1 | 3 | | |
| L46L | CTG>CTA | | | 1 | |
| S47N | AGT>AAT | 1 | | | |
| G48D | GGT>GAT | 1 | | | |
| **Total** | | 42 | 9 | 10 | 6(5)* |

* One large pilomatricoma demonstrated biclonal mutations

non-syndromic pilomatricoma. Entries in the COSMIC database covering more than 7000 *CTNNB1* mutations confirm that mutations at codons 33 and 37 are equally represented in human cancers and further show that the most mutated codons of *CTNNB1* are 41 and 45 (https://cancer.sanger.ac.uk/cosmic/gene/analysis?ln=CTNNB1). Only one other insertion mutation encompassing codon 34 is listed in the COSMIC database but it is different from the mutation detected in our patient and does not represent a DNA-duplication.

We checked whether the four observed base substitution mutations: T[C>T]T, T[C>G]T, T[C>A]C, A[C>T]C would be suggestive of one of the 30 published mutations signatures (https://cancer.sanger.ac.uk/cosmic/signatures); however, the substitutions did not represent only one pattern but were found to predominate in patterns 2, 13, 24 and 12. Interestingly, these four patterns are assumed to display a transcriptional strand bias.

## Analysis of CTNNB1 expression and of genes downstream of CTNNB1

CTNNB1 immunohistochemistry was performed on one pilomatricoma. CTNNB1 expression was strongly upregulated as it has been described before in non-syndromic pilomatricoma [28]. In the pilomatricoma, expression was limited to the matrix cells (Fig 2A). Protein expression of genes *CCND1* and *c-myc* which are downstream of CTNNB1 in the WNT-pathway could be demonstrated as well, but protein expression was relatively weak as expected for a benign tumor (Fig 2B and 2C). These findings are consistent with the assumption that the CTNNB1 mutations represent the driving event in DM1-associated pilomatricoma.

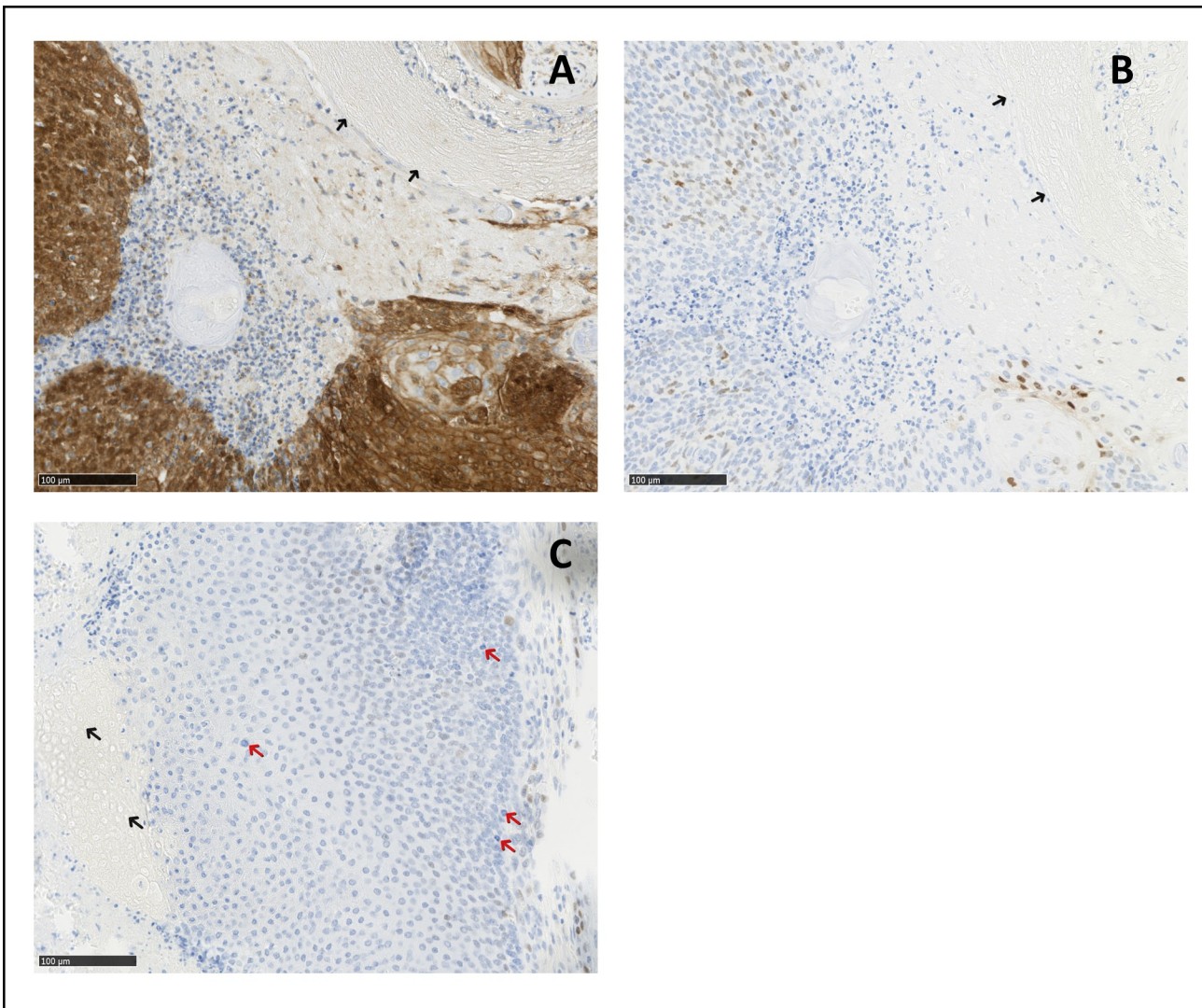

**Fig 2. Immunohistochemistry of CTNNB1, CCND1 and c-myc in the pilomatricoma p.(G34dup).** A: The stain for CTNNB1 shows a strong nuclear reaction in the tumor cells. Note the negative staining of the shadow-cells (arrows). B: CCND1. Only a minority of tumor cells show a distinct positive reaction for cyclin D1. Note the negative staining of the shadow-cells (arrows). C: C-myc. The image demonstrates a faint nuclear staining in a small percentage of cells. Note the negative staining of the shadow-cells (arrows) and the mitotic figures in the tumor cells (red arrows).

## Analysis of microsatellite instability and expression of mismatch repair proteins

As multiple pilomatricomas with *CTNNB1* mutations have been described in patients with *PMS2* germline mutations, we analyzed the stability of microsatellite DNAs BAT25, BAT26, D2S123, D5S354 and D17S250 in two pilomatricomas of the patient (G34dup, T41I) and in the pilomatrical carcinoma. All tested microsatellites remained stable compared to lymphocyte DNA. The pilomatricoma with the duplication mutation was further analyzed at microsatellite markers BAT40, D10S197, NR21, NR22 and NR24 without demonstration of microsatellite instability.

Expression of mismatch repair proteins was analyzed by immunohistochemistry staining proteins MLH1, MSH2, MSH6 and PMS2 in two pilomatricomas (G34dup, T41I) and a strong

expression of all four proteins was detected in all samples. In the pilomatricomas, expression of mismatch repair proteins was restricted to the basaloid cells (Fig 1F).

## NGS-analysis of 161 cancer-related genes

In addition, 161 cancer-related genes were screened for mutations using the Oncomine Comprehensive Assay v3 with DNA and RNA from the pilomatricoma with the duplication mutation Glu34dup (allele frequency: 25% [203x/823x]) and from the pilomatrical carcinoma. NGS sequencing confirmed the presence of the individual CTNNB1 mutations. NGS of genes *MLH1*, *MSH2*, *MSH6*, *PMS2* and *POLE* did neither reveal germline or somatic gene mutations in the patient's pilomatricoma (G34dup.), nor in the pilomatrical carcinoma or in blood lymphocytes. However, in both samples as well as in the blood DNA of the patient we could detect a pathogenic germline mutation in heterozygous state within the *ATM* gene (ATM: NM_000051.3; c.8494C>T; p. Arg2832Cys, blood: allele frequency: 50% [280x/560x]). Besides known polymorphisms, no other pathogenic mutations could be found in the pilomatricoma, the pilomatrical carcinoma and the patient's blood DNA. CNV-analysis based on NGS-data did not reveal chromosomal instability in the pilomatricoma and in the pilomatrical cancer. The raw sequencing data of the pilomatrical carcinoma are deposited in the National Center for Biotechnology Information Sequence Read Archive (https://www.ncbi.nlm.nih.gov/sra); accession number PRJNA603431, SubmissionID SUB6884814.

## Discussion

Up to now it is not known which molecular mechanisms might be responsible for the occurrence of multiple pilomatricomas in patients with DM1. Likewise, the mechanism which might account for the enhanced cancer risk in DM1 is unknown. We report the first mutation analysis of the *CTNNB1* gene in multiple pilomatricomas and in one pilomatrical carcinoma obtained from a single patient with molecular proven DM1. The presence of somatically acquired mutations of exon 3 of the *CTNNB1* gene could be demonstrated. Moreover, five different *CTNNB1* mutations could be demonstrated in these tumors (S33C, S33F, G34V, T41I, G34dup) which evidences that mutations arose somatically and independently in each tumor.

The detection of different somatic *CTNNB* mutations in different pilomatricomas and in the pilomatrical carcinoma as well as the fact that the patient developed 10 pilomatricomas and one pilomatrical carcinoma within 4 years, strongly suggests that the patient displays a hypermutation phenotype. The distribution of mutations detected in the tumors of the patient seems to differ slightly from the mutation distribution displayed by non-syndromic pilomatricomas and pilomatrical carcinomas as no mutations were found in codon 37; however, the number of sequenced tumors in the patient is too low for any statistical proof (Table 1).

The degree of genetic instability present in DM1 patients most likely varies considerably. Some DM1 patients develop multiple pilomatricomas which suggests a greatly enhanced mutation rate at the *CTNNB* gene in these patients, but DM1 patients with pilomatricomas still seem to represent only a minority of all DM1 patients. This could suggest that an additional mutated gene or a polymorphism in one or several genes act as modifier of a putative hypermutation phenotype. Although one mutation detected in the patient's pilomatricoma involved a small duplication which might suggest that mismatch repair is reduced in the patient, analysis of microsatellite size within two pilomatricomas and the pilomatrical carcinoma of the patient did not reveal microsatellite instability. In addition, NGS of genes *MLH1*, *MSH2*, *MSH6*, *PMS2* and *POLE* did not reveal germline or somatic mutations in the patient's pilomatricoma (G34dup.), the pilomatrical carcinoma as well as in blood lymphocytes. Likewise, analysis of expression of DNA mismatch repair proteins did not reveal a defect within the MMR pathway in the studied pilomatricomas.

In order to detect additional gene mutations which might modify genetic instability we performed NGS analysis on 161 cancer-related genes with tumor material of the pilomatricoma with the G34 duplication and of the pilomatrical carcinoma and compared the result with the patient's blood.

The only additional mutation which could be detected by NGS was the heterozygous germline mutation c.8494C>T; p.(Arg2832Cys) within the ataxia telangiectasia mutated gene (*ATM*). Biallelic inactivation of *ATM* induces Ataxia telangiectasia (A-T) which is an autosomal recessive disorder with cerebellar degeneration, telangiectasia, immunodeficiency and cancer susceptibility [33]. A-T-cells display radiation sensitivity due to a defect in repair of DNA double strand breaks. Cancer spectrum of A-T does not overlap with cancers found in DM1 patients. Moreover, the wild type allele of *ATM* was retained in the pilomatricoma as well as in the pilomatrical carcinoma which suggests that biallelic functional inactivation of ATM did not play a role in the development of pilomatricoma and pilomatrical carcinoma. Nevertheless, the *ATM* missense mutation c.8494C>T; p.(Arg2832Cys) has been associated with an increased cancer risk even in heterozygous carriers [33], therefore a disease modifying role in DM1-associated cancer susceptibility might not be ruled out completely and other patients with DM1 and pilomatricomas should be screened for defects in cancer-driving genes. Unfortunately, no NGS data on non-syndromic pilomatricomas are available which would allow a comparison between the two entities.

Although the molecular mechanisms responsible for the hypermutation phenotype remain unexplained in the described patient, the multiple occurrence of pilomatricomas with individual somatic *CTNNB1* mutations suggests some characteristics of the putative genetic defect:

The *CTNNB1* gene as well as the hair matrix cells seem to be preferentially targeted by the unknown genetic defect as pilomatricoma is a rare benign neoplasm and other potential CTNNB1 driven neoplasms do not seem to be more frequent in DM1 patients with the exception of endometrial cancer. According to the COSMIC database, *CTNNB1* mutations have been detected at more than 10% frequency in neoplasms of pituitary (37%), soft tissue (36%), liver (22%), endometrium (18%), adrenal gland 13% and small intestine 12% (only entries with more than 100 sequenced samples):. Tissue distribution of DMPK-RNA-expression might represent a modifying factor as *DMPK*-RNA seems to be present in cycling keratinocytes, in hair follicles as well as in endometrial tissue [34] (see also The Human Protein Atlas, https://www.proteinatlas.org/ENSG00000104936-DMPK/tissue). Simultaneous transcription of the mutated *DMPK* gene and the *CTNNB1* gene in cycling hair follicles might be responsible for tissue and gene specificity and could be an explanation for the putative mutation signatures detected in the patient's tumor specimens which suggest a transcriptional mutational bias. In other cancers, co-transcription has been proposed as a mechanism responsible for gene fusions [35]. Simultaneous transcription of *DMPK* and *CTNNB1* resulting in a defect of transcription coupled DNA repair at the *CTMNB1* gene could further provide an explanation why no additional mutation could be detected within the other 160 cancer-related genes which were screened by NGS even though the patient obviously displays a hypermutation phenotype. On the other hand, the used panel only encompasses about 0.4 Mb cumulative target size and it is known that several cancer types show less than 1 mutation per Mb [36].

The putative hypermutability by toxic *DMPK* gene-derived RNA might be induced through defective splicing of mRNA of genes with proofreading function and of genes implicated in DNA replication. Alternatively, putative hypermutability might result from a direct interfering effect of toxic RNA on proofreading or replication. Fig 3 exemplifies these two hypotheses.

The observation that multiple pilomatricomas with *CTNNB1* mutations have been observed in patients with constitutive mismatch repair deficiency (CMMR-D) associated with *PMS2* germline mutations [22] and that *CTNNB1* mutations are frequent in colon cancers of

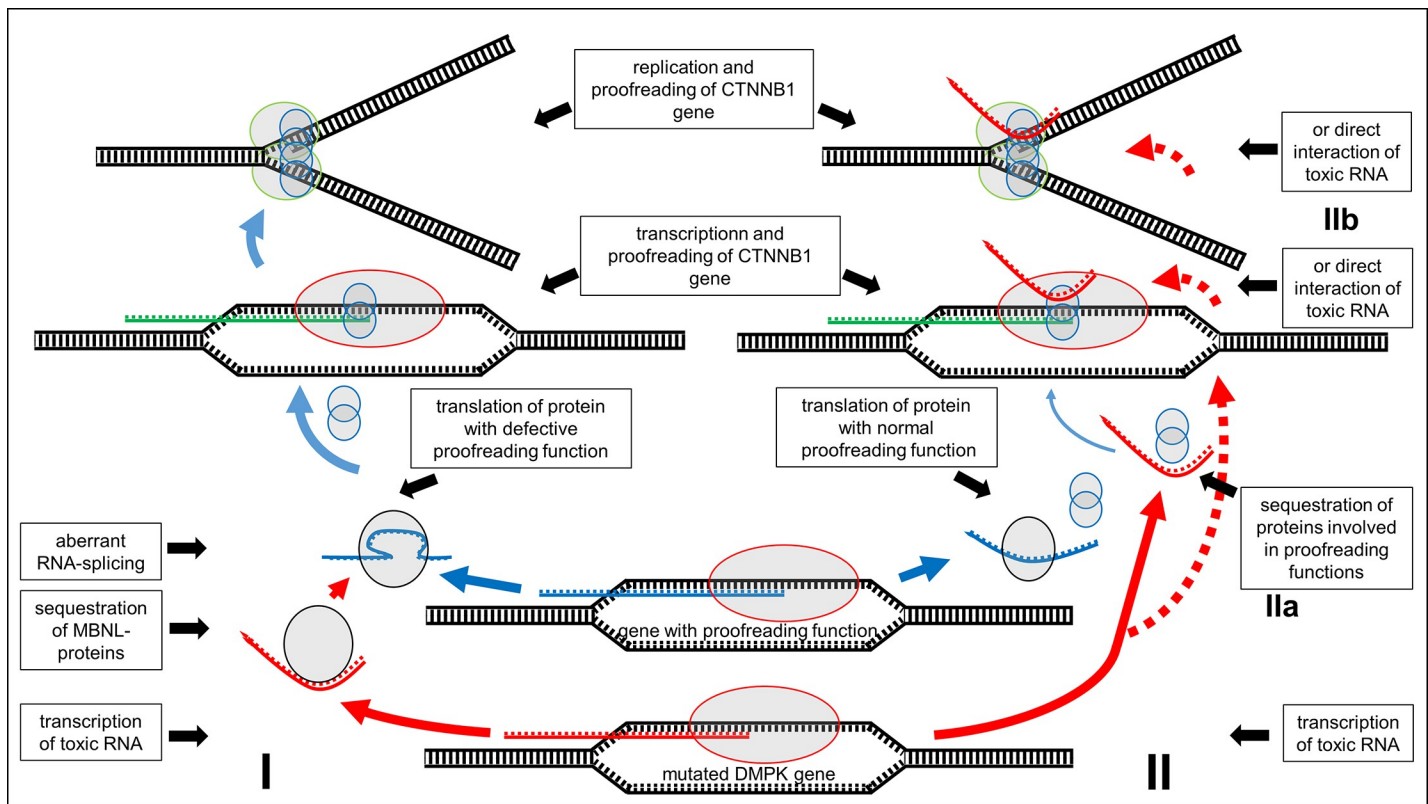

**Fig 3. Two hypotheses on interaction of toxic RNA from mutated *DMPK* gene.** I: Toxic RNA interferes with splicing of RNA from genes with proofreading function. Defective proteins enhance mutation rate during transcription and replication of *CTNNB1*. II: Toxic RNA sequesters proteins involved in proofreading (IIa) or interferes directly at the site of transcription or replication of *CTNNB1* (IIb) and thereby enhances mutation rate during transcription and replication.

HNPCC patients with *MLH1* or *MSH2* germline mutations suggested that the DNA mismatch repair mechanisms might play a role in *CTNNB1* mutation susceptibility. Interestingly, analysis of CMMR-D-associated pilomatricomas did not reveal microsatellite instability with markers BAT-26, BAT-25, BAT-40, D2S123, D5S346, D17S250, TGFbRII, D17S787, D18S58 and D18S69 despite mutations of *PMS2* [22]. This might indicate that lack of microsatellite instability in pilomatricomas may not rule out a causative role of DNA mismatch repair in the enhanced mutation rate of the *CTNNB1* gene. Most importantly, while defects of DNA mismatch repair proteins lead to microsatellite instability, it seems that the expansion of trinucleotide repeats is linked to overexpression of the mismatch repair proteins MSH2, MSH3 or PMS2 [7,9,34]. DNA mismatch repair proteins do not only play a role in post replication DNA mismatch repair but also seem to be implicated in double strand break repair, transcription-coupled repair and nucleotide excision repair [37]. Alterations of the DNA mismatch repair proteins might therefore still be responsible for the observed enhanced mutation rate of the CTNNB1 gene in the pilomatricomas of the DM1 patient. A BRCA1-associated genome surveillance complex (BASC) has been hypothesized, which contains BRCA1, MSH2, MSH6, MLH1, ATM, BLM, PMS2 and the RAD50-MRE11-NBS1 protein [38]. The presence of ATM in BASC could hint to a link between the enhanced mutation rate of *CTNNB1* in the analyzed pilomatricomas and the detected *ATM* germline mutation.

In conclusion, molecular analysis of four pilomatricomas and one pilomatrical carcinoma in a patient with myotonic dystrophy type 1 demonstrated that the patient displayed hypermutability within his hair matrix cells targeting the catenin-β gene which suggests a tissue and

gene restricted hypermutation phenotype associated with DM1. More molecular research on DM1 cancer predisposition will have to be performed in order to identify the mechanisms responsible for putative hypermutability in DM1 patients.

## Materials and methods

### Ethics statement

The University ethics committee (Ethik-Kommission an der Medizinischen Fakultät der RWTH Aachen) approved this research (EK-314-19, written consent).

Genetic analyses were performed with written consent of the patient and germline mutation analysis was undertaken after genetic counseling as required by German law. The patient gave written consent for publication.

### Patient's characteristics

The male patient was 39-year-old when he first presented with two pilomatricomas located on the scalp and on the left elbow. Until the age of 43 he developed 8 additional pilomatricomas located on the scalp as well as one pilomatrical carcinoma of the scalp. The clinical diagnosis of myotonic dystrophy was first assumed at the age of 27 when he demonstrated muscle myotonia, sleep apnea, bilateral ptosis, mild cataract and characteristic changes of the electromyogram. Besides pilomatricomas, dermatologic examination of the patient revealed multiple (>50) melanocytic nevi as previously described in myotonic dystrophy patients [39], frontal baldness as well as one neurofibroma located on the chest. Radiologic staging for pilomatrical carcinoma revealed a 4 cm large left-sided thyroid nodule which was benign according to fine needle biopsy. The molecular diagnosis of DM1 was confirmed at that time. No symptoms of DM1 were present in both parents, in his brother and sister, as well as in his sister's three children. A molecular analysis of the *DMPK* gene was not performed in the patient's relatives. His grandmother died from an unknown cancer; an uncle died at the age of 63 from prostate cancer while another uncle died from colon cancer at the age of 50.

Five pilomatricomas and one pilomatrical carcinoma were analyzed in this study. One pilomatricoma was excised from the left elbow and was 1.3 cm in size. The other four pilomatricomas were located on the scalp and ranged in size from 0.9 cm to 1.5 cm. The pilomatrical carcinoma was located on the scalp and measured 2.9 cm in diameter.

### Analysis of CTG repeat expansions

CTG repeat expansion within the *DMPK* gene was determined by PCR and Southern blot analysis as previously described [3,4].

### Analysis of catenin-β gene mutations

Five pilomatricomas and one pilomatrical carcinoma were analyzed after annotation of the regions of interest. Percentage of tumor cells (tumor cell content) was estimated as the percentage of tumor cell nuclei in relation to all other cell nuclei (e.g. stromal, inflammatory, epithelial,. . .). Tumor tissue was manually microdissected from the slides and FFPE-DNA and RNA was isolated with the Maxwell system (Promega) according the manufacturer's protocol. DNA from lymphocytes was isolated by salting out method.

Tumor cell fraction was at least >20% in all cases. All cases were sequenced by sanger sequencing after amplification of exon 3 of the *CTNNB1* gene by PCR (reference genome NCBI, hg19/NM_ 001904.3) [40]. Sequence analysis was performed with JSI SeqPilot Software (SeqPatient module).

### Next generation sequencing analysis of pilomatricoma and pilomatrical carcinoma

Additionally, next generation sequencing (NGS) was performed with the Ampliseq Comprehensive Assay v3 for Illumina with DNA and RNA from two patients' tumor samples (pilomatrical carcinoma, tumor cell fraction >80%, one pilomatricoma, tumor cell fraction >40%) and blood according to the manufacturer's instructions (reference genome NCBI, hg19). Libraries were sequenced on the NextSeq or MiSeq platform (Ilumina) respectively. Bam files were generated with the DNA and RNA Amplicon Module (Illumina). Fusion calling and expression analysis of the RNA was also performed with the RNA Amplicon module, DNA variant analysis was performed with the JSI SeqPilot Software (SeqNext module), variants with an allele frequency >10% were further analyzed. Variants with an allele frequency of >1% in the normal population according to gnomAD (https://gnomad.broadinstitute.org/) were considered benign polymorphisms. Copy number variation (CNV) analysis was performed with an in-house algorithm (manuscript in preparation).

### Analysis of microsatellite instability and expression of mismatch repair proteins

Extracted DNA for NGS analysis was also used to perform MSI testing. DNA tumor samples and corresponding normal tissue were PCR-amplified with the Bethesda marker panel (BAT25, BAT26, D2S123, D5S354 and D17S250) and in one sample also with markers BAT40, D10S197, NR21, NR22 and NR24 in a multiplex-PCR with fluorescence-tagged primers. The fragment sizes were displayed by co-electrophoresis using the Genetic Analyzer 3500 capillary sequencer (Thermofisher). Fragment length analysis was performed using Genemapper Software 5 (Thermofisher).

### Analysis of expression of DNA mismatch repair proteins, CTNNB1, CCND1 and c-myc

Immunohistochemical staining of DNA mismatch repair proteins as well as of proteins CTNNB1, CCND1 (cyclin D1) and c-myc in tumor tissue was performed as follows: 3–4 μm slides were cut and stained for MLH1 (Monoclonal Mouse Anti-Human; Clone ES05, dilution 1 : 10, Dako), MSH2 (Monoclonal Mouse Anti-Human; Clone G219-1129, dilution 1 : 200, BD Biosciences), MSH6 (Monoclonal Rabbit Anti-Human; Clone EP49, dilution 1 : 1000 Dako), PMS2 (Monoclonal Mouse Anti-Human; Clone A 16–4, dilution 1 : 100, BD Biosciences) to assess the reactivity in the nuclei of tumor cells as well as for CTNNB1 (Monoclonal Mouse Anti-Human; clone β-Catenin-1, ready to use, Dako), CCND1 (Monoclonal Rabbit Anti-Human; Clone EP12, ready to use, Dako) and c-myc (Monoclonal Rabbit Anti-Human; Clone Y69, 1:100 dilution, Abcam). Immunostains were developed according to an antigen retrieval treatment using a detection system suitable for the Dako Autostainer Link 48 (EnVision FLEX, Dako).

## Acknowledgments

We would like to thank Inge Losen, Department of Dermatology, RWTH Aachen University, for her technical support.

## Author Contributions

**Conceptualization:** Albert Rübben.

**Data curation:** Nadina Ortiz-Brüchle.

**Formal analysis:** Thomas Eggermann.

**Investigation:** Albert Rübben, Renate Ursula Wahl, Thomas Eggermann, Edgar Dahl, Nadina Ortiz-Brüchle, Claudio Cacchi.

**Methodology:** Thomas Eggermann, Edgar Dahl, Nadina Ortiz-Brüchle, Claudio Cacchi.

**Project administration:** Albert Rübben, Renate Ursula Wahl, Edgar Dahl.

**Resources:** Edgar Dahl.

**Supervision:** Edgar Dahl.

**Validation:** Thomas Eggermann.

**Visualization:** Nadina Ortiz-Brüchle, Claudio Cacchi.

**Writing – original draft:** Albert Rübben.

**Writing – review & editing:** Renate Ursula Wahl, Thomas Eggermann, Edgar Dahl, Nadina Ortiz-Brüchle, Claudio Cacchi.

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
