## [Decision Letter · Decision Letter 0]

18 Dec 2019

PONE-D-19-32249

Mutation analysis of multiple pilomatricomas in a patient with myotonic dystrophy type 1 suggests a DM1-associated hypermutation phenotype

PLOS ONE

Dear Prof. Dr. Rübben,

Thank you for submitting your manuscript to PLOS ONE. After careful consideration, we feel that it has merit but does not fully meet PLOS ONE’s publication criteria as it currently stands. Therefore, we invite you to submit a revised version of the manuscript that addresses the points raised during the review process.

We would appreciate receiving your revised manuscript by Feb 01 2020 11:59PM. To enhance the reproducibility of your results, we recommend that if applicable you deposit your laboratory protocols in protocols.io, where a protocol can be assigned its own identifier (DOI) such that it can be cited independently in the future. For instructions see: http://journals.plos.org/plosone/s/submission-guidelines#loc-laboratory-protocols

We look forward to receiving your revised manuscript.

Kind regards,

Ruben Artero, Ph.D.

Academic Editor

PLOS ONE

Journal Requirements:

1.

Additional Editor Comments (if provided):

The manuscript is of interest to the journal but reviewers raise a number of technical concerns that require a detailed response by the authors. Reviewer 1 requests some experiments about confirming activation of the Wnt pathway, which seem feasible, and are therefore requested before a revised version of the manuscript is submitted.

Reviewers' comments:

Reviewer's Responses to Questions

**Comments to the Author**

1. Is the manuscript technically sound, and do the data support the conclusions?

Reviewer #1: Yes

Reviewer #2: Yes

Reviewer #3: Yes

2. Has the statistical analysis been performed appropriately and rigorously? 

Reviewer #1: Yes

Reviewer #2: Yes

Reviewer #3: Yes

3. Have the authors made all data underlying the findings in their manuscript fully available?

Reviewer #1: Yes

Reviewer #2: No

Reviewer #3: Yes

4. Is the manuscript presented in an intelligible fashion and written in standard English?

Reviewer #1: Yes

Reviewer #2: Yes

Reviewer #3: Yes

5. Review Comments to the Author

Reviewer #1: We have read the paper presented by Rübben et al. focused in mutation analysis of a patient with myotonic dystrophy with multiple pilomatrocomas. The paper is well presented and data are supported by the experimental results. Nevertheless, regarding the conclusions it is worth to made several considerations:

(i) The authors should tone down their conclusions, since samples are obtained from only individual, and focus more on presenting their results rather than confronting previous ideas and or hypothesis.

(ii) Authors should check whether Wnt pathway is activated in the different tumors checking expression of B-catenin and APC by immunohistochemistry and downstream targets such c-myc or cyclin D among others.

(iii) Results from NGS only found a mutation in ATM gene. I would have expect higher percentage. Do the authors have an explanation for this low number.

(iv) Authors should indicate whether pilomatricomas and non syndromic pilomatricoma present mutations within the genes checked in NGS studies and discuss the differences with their study.

Reviewer #2: This is a very interesting finding. I recommend to accept the manuscript when following points are addressed:

The CTNNB1 mutations the authors have identified in the cancers of the myotonic dystrophy patients have been shown to sensitize tumor cells to TTK kinase inhibitors (Mol. Cancer Ther. 16(11) 2609-17). Do the authors believe that treatment with TTK inhibitors could be a viable option for these patients?

Please provide more technical detail on how the relative fraction of tumor cells was determined in the samples that were sequenced in the Methods section.

Please provide more details on the results of the sequence analysis, more specifically, the % of reads that were harboring the mutant vs. the wild-type CTNNB1 gene.

Reviewer #3: I am very happy to have an opportunity to review this interesting paper. The authors investigated somatic mutation of the CTNNB1 gene in multiple pilomatricomas and a pilomatrical carcinoma developed in a patient with DM1, revealed no microsatellite instability, and ruled out mutations in genes for mismatch repair proteins. This manuscript is well-written and very interesting. I have some comments and questions, which could help you revise this manuscript.

1) I would like to know the clinical information of each tumor (size, location, etc). Because multiple pilomatricoma in DM1 patients sometimes get larger than non-syndromatic soliary pilomatricoma.

2) Your hypothesis on interaction of toxic RNA from mutated DMPK gene is really interesting. I was just wondering if co-transation of DMPK gene and CTNNB1 gene has already been evidenced in the literature or not? Or is it just a hypothesis? I was not able to find any reference about that in this paper.

6. PLOS authors have the option to publish the peer review history of their article (what does this mean?). If published, this will include your full peer review and any attached files.

Reviewer #1: No

Reviewer #2: No

Reviewer #3: No

---

## [Author Response · Author response to Decision Letter 0]

3 Feb 2020

1) Have the authors made all data underlying the findings in their manuscript fully available? Reviewer #2: No

In order to share the raw NGS-data we have uploaded the data under: 

SRA: https://www.ncbi.nlm.nih.gov/sra SubmissionID: SUB6884814 BioProject ID: PRJNA603431 and included this information in the revised manuscript.

2) Reviewer #1: (i) The authors should tone down their conclusions, since samples are obtained from only individual, and focus more on presenting their results rather than confronting previous ideas and or hypothesis.

We agree to the reviewer and have soften author summary, introduction and discussion to this regard.

3) Reviewer #1: (ii) Authors should check whether Wnt pathway is activated in the different tumors checking expression of B-catenin and APC by immunohistochemistry and downstream targets such c-myc or cyclin D among others.

We have introduced immunohistochemistry for expression of CTNNB1, c-myc and cyclin D1 and could demonstrate that CTNNB1 is highly expressed in the pilomatricoma which further supports a causative role of CTNNB1 mutations. This has already been described before in non-syndromic pilomatricoma. Expression of c-myc and cyclin D could be detected as well, but at a lower level. Nevertheless, lower expression of c-myc and cyclin D is consistent with pilomatricoma being a benign and very slowly growing tumor. Absence of expression of c-myc and cyclin D in shadow cells which do not express CTNNB1 further supports the assumption that WNT-CTNNB1-pathway plays a crucial role in pilomatricoma.

4) Reviewer #1: (iii) Results from NGS only found a mutation in ATM gene. I would have expected higher percentage. Do the authors have an explanation for this low number.

Indeed, this is a low number of detected mutations, however, in our experience this is not a totally unexpected finding for a panel that encompasses only about 0.4 Mb cumulative target size. The finding is also in line with published data on large tumor mutational burden studies. E.g. Chalmers et al. analyzed 100.000 human cancer genomes and showed that several cancer types show less than 1 mutation/Mb1.

Moreover, our assumption is that there might be a transcriptional bias which favors mutation acquisition in CTNNB1 but does not lead to a high mutation load in the whole genome.

1Chalmers ZR, Connelly CF, Fabrizio D, Gay L, Ali SM, Ennis R, Schrock A, Campbell B, Shlien A, Chmielecki J, Huang F, He Y, Sun J, Tabori U, Kennedy M, Lieber DS, Roels S, White J, Otto GA, Ross JS, Garraway L, Miller VA, Stephens PJ, Frampton GM. Analysis of 100,000 human cancer genomes reveals the landscape of tumor mutational burden. Genome Med. 2017 Apr 19;9(1):34.

5) Reviewer #1: (iv) Authors should indicate whether pilomatricomas and non-syndromic pilomatricoma present mutations within the genes checked in NGS studies and discuss the differences with their study.

This would certainly be a very interesting scientific question that needs to be answered. However, to the best of our knowledge, no NGS data on pilomatricomas have been published to date.

6) Reviewer #2: The CTNNB1 mutations the authors have identified in the cancers of the myotonic dystrophy patients have been shown to sensitize tumor cells to TTK kinase inhibitors (Mol. Cancer Ther. 16(11) 2609-17). Do the authors believe that treatment with TTK inhibitors could be a viable option for these patients?

The correlation between overexpression of the assembly checkpoint kinase TTK (Mps1) and CTNNB1 mutations is not understood. It could be that Mps1 overexpression favors the occurrence of CTNNB1 mutations or that CTNNB1 overexpression by activating mutations enhances Mps1 expression. In the first case, we would not expect a role of TTK inhibitors in treating tumors of DM1 patients as we assume that CTNNB1 mutations are due to simultaneous transcription of CTNNB1 and the mutated DMPK gene. In the latter case, we could expect a role in treatment if the tumors of DM1 patients would acquire aneuploidy due to Mps1 overexpression, but this would only be the case in malignant tumors and not in pilomatricomas. We have not included this discussion in the revised manuscript.

7) Reviewer #2: Please provide more technical detail on how the relative fraction of tumor cells was determined in the samples that were sequenced in the Methods section. Please provide more details on the results of the sequence analysis, more specifically, the % of reads that were harboring the mutant vs. the wild-type CTNNB1 gene.

We have included the requested information in the manuscript.

8) Reviewer #3: I would like to know the clinical information of each tumor (size, location, etc.). Because multiple pilomatricoma in DM1 patients sometimes get larger than non-syndromic solitary pilomatricoma.

We have included the requested information in the manuscript.

9) Reviewer #3: Your hypothesis on interaction of toxic RNA from mutated DMPK gene is really interesting. I was just wondering if co-translation of DMPK gene and CTNNB1 gene has already been evidenced in the literature or not? Or is it just a hypothesis? I was not able to find any reference about that in this paper.

This is just our hypothesis. We replaced co-translation by simultaneous transcription in the manuscript as the putative interaction should already take place at the stage of transcription as described in figure 3 (2). We did not use the term co-transcription as this term is also used for specific forms of post transcriptional modifications. Nevertheless, co-transcription in the sense of simultaneous transcription has been proposed as a mechanism for gene fusions: Wright RL and Vaughan AT. A systematic description of MLL fusion gene formation.Crit Rev Oncol Hematol 2014 - Review. PMID 24787275. We have included this reference.

---

## [Decision Letter · Decision Letter 1]

20 Feb 2020

Mutation analysis of multiple pilomatricomas in a patient with myotonic dystrophy type 1 suggests a DM1-associated hypermutation phenotype

PONE-D-19-32249R1

Dear Dr. Rübben,

We are pleased to inform you that your manuscript has been judged scientifically suitable for publication and will be formally accepted for publication once it complies with all outstanding technical requirements.

With kind regards,

Ruben Artero, Ph.D.

Academic Editor

PLOS ONE

Additional Editor Comments (optional):

Reviewers' comments:

Reviewer's Responses to Questions

**Comments to the Author**

1. If the authors have adequately addressed your comments raised in a previous round of review and you feel that this manuscript is now acceptable for publication, you may indicate that here to bypass the “Comments to the Author” section, enter your conflict of interest statement in the “Confidential to Editor” section, and submit your "Accept" recommendation.

Reviewer #1: All comments have been addressed

Reviewer #3: All comments have been addressed

2. Is the manuscript technically sound, and do the data support the conclusions?

Reviewer #1: (No Response)

Reviewer #3: (No Response)

3. Has the statistical analysis been performed appropriately and rigorously? 

Reviewer #1: (No Response)

Reviewer #3: (No Response)

4. Have the authors made all data underlying the findings in their manuscript fully available?

Reviewer #1: (No Response)

Reviewer #3: (No Response)

5. Is the manuscript presented in an intelligible fashion and written in standard English?

Reviewer #1: (No Response)

Reviewer #3: (No Response)

6. Review Comments to the Author

Reviewer #1: (No Response)

Reviewer #3: The author responded to all comments and questions adequately.

I have no comments any more.

Great paper.

7. PLOS authors have the option to publish the peer review history of their article (what does this mean?). If published, this will include your full peer review and any attached files.

Reviewer #1: No

Reviewer #3: No

---

## [Editor Report · Acceptance letter]

25 Feb 2020

PONE-D-19-32249R1 

Mutation analysis of multiple pilomatricomas in a patient with myotonic dystrophy type 1 suggests a DM1-associated hypermutation phenotype 

Dear Dr. Rübben:

I am pleased to inform you that your manuscript has been deemed suitable for publication in PLOS ONE. Congratulations! Your manuscript is now with our production department. 

With kind regards,

on behalf of

Dr. Ruben Artero 

Academic Editor

PLOS ONE